# Enhanced Communications on Satellite-Based IoT Systems to Support Maritime Transportation Services

**DOI:** 10.3390/s22176450

**Published:** 2022-08-26

**Authors:** Victor Monzon Baeza, Flor Ortiz, Samuel Herrero Garcia, Eva Lagunas

**Affiliations:** 1SIGCOM Group, Interdisciplinary Centre for Security Reliability and Trust (SnT), University of Luxembourg, 1855 Luxembourg, Luxembourg; 2Telefonica Ronda de la Comunicacion, C4, 28050 Madrid, Spain

**Keywords:** zigbee, maritime transport and logistic services, satellite-based Internet of Things

## Abstract

Maritime transport has become important due to its ability to internationally unite all continents. In turn, during the last two years, we have observed that the increase of consumer goods has resulted in global shipping deadlocks. In addition, the future goes through the role of ports and efficiency in maritime transport to decarbonize its impact on the environment. In order to improve the economy and people’s lives, in this work, we propose to enhance services offered in maritime logistics. To do this, a communications system is designed on the deck of ships to transmit data through a constellation of satellites using interconnected smart devices based on IoT. Among the services, we highlight the monitoring and tracking of refrigerated containers, the transmission of geolocation data from Global Positioning System (GPS), and security through the Automatic Identification System (AIS). This information will be used for a fleet of ships to make better decisions and help guarantee the status of the cargo and maritime safety on the routes. The system design, network dimensioning, and a communications protocol for decision-making will be presented.

## 1. Introduction

### 1.1. Background

Today, emerging communications technologies are contributing to the evolution of cities toward Smart Cities. This favors the sustainability of the city in terms of resources such as relation between energy savings and well-being of citizens. Within cities, different economic sectors or verticals such as the energy sector, health, education, infrastructure, and transportation are being improved by the use of smart communications (Zdraveski et al. in [1]).

There are different smart applications as a solution to different services within the Smart City (Zafeiriou et al. in [2]). These smart solutions are articulated through connected modular and scalable systems that facilitate the control and management of different environments in a city. They can be cataloged in different blocks as follows:**Smart Environment**: focused on environmental management systems to improve energy efficiency and the quality of the environment in cities.**Smart Mobility**: providing intelligent monitoring systems for the transport and mobility of a city and its surroundings to be more efficient and reduce the carbon footprint.**Smart Living**: intelligent fire detection systems, video surveillance, and air conditioning to improve the quality of life of citizens.**Smart People**: smart solutions for citizens to integrate intelligent communication between the city and its citizens (see interactive maps, citizen apps, and social Wi-Fi).

Among the emerging technologies used in Smart Cities are the Internet of Thing (IoT) and Big Data (Talebkhah et al. in [3]). Through IoT, the city has an intelligent network of connected objects and machines that transmit data using wireless technology protocols such as Long Range (LoRa), Zigbee, and Bluetooth, among others, and Message Queuing Telemetry Transport (MQTT) and Web Application Messaging Protocol (WAMP) for the cloud environment. Supported by Big Data, Smart Cities use these data to implement measures that improve the quality of life of the inhabitants.

The technological transformation of cities is posing several technical challenges. On one hand, climate change is one of the main challenges to be addressed in the coming years within the evolution of Smart Cities. 11% of global greenhouse gas emissions come from the logistics sector, with 3% coming from maritime transport. For this reason, we must pay attention to coastal ports and efficiency in maritime transport, with the aim of decarbonizing their impact on the environment Ahwazi et al. in [4].

On the other hand, recently, the COVID-19 pandemic has opened another challenge that must be faced since it has led to a change in the consumption habit in society Koelle et al. in [5]. During these years, there has been greater demand from the transport sector, especially in the maritime case. About 80% of the things we consume are transported by sea. The transport of containers through shipping companies has fostered a change in consumption and purchasing patterns. As a result of the increase in the demand for goods and an unprecedented shortage of empty containers, maritime freight rates have skyrocketed by up to 328%, while the demand for air transport fell by 90% (Koelle et al. in [5]).

In just 11 months, the average shipping price of the standard container used in maritime transport, known as twenty-foot equivalent unit (TEU), and the most common in international trade has gone from 1103 dollars to 3785 dollars (Koelle et al. in [5]). This increase in maritime logistics transport clearly impacts the final consumer. The delays caused by the shortage of empty containers also affect the final costs, making container movements even more expensive and increasing prices. In the order of magnitude of this problem, it can be pointed out that the current level of freight costs is equivalent to values that oscillate between 0.35% of the retail value in the case of high-value clothing and 63.55% in the case of high volume and low value in furniture Koelle et al. in [5].

Entering the type of merchandise transported, we find the problem that certain products require special treatment or greater supervision, where any unforeseen event can spoil the merchandise, resulting in large economic losses or directly impacting the health of the citizen. This is the case of certain foods or medicines that need special temperature conditions during the shipping process. During the pandemic, the amount of transported medicines increased, 80% of the merchandise transported by sea. Faced with this situation of increasing maritime demand, monitoring the cold chain on the high seas and, consequently, acting in the face of problems that arise is a challenge to be tackled. To do this, satellite networks breakthrough with their innate global coverage capability as a key candidate to offer coverage on the high seas.

This work focuses on finding a solution for various services offered in maritime transport that resolve the problems introduced within this sector, responding to the challenges posed. Within the Smart Cities applications, this work will be included as solutions for Smart Ports (*Smart Environment environment*) and Smart Transport (*Smart Mobility environment*).

### 1.2. Related Work

In this section, we review several works related to intelligent transportation (IT), its evolution, as well as IoT technology and satellite communications (SatCom) within the maritime transport sector (MTS) as topics related to the proposal presented in this work.

In the IT sector, the focus has been mostly on ground transportation. Many systems and proposals for vehicular communications have been developed to advance autonomous driving. To provide a higher level of safety for both driving and the driver. These systems make cities more sustainable with the environment by also controlling emissions through traffic management (Cao et al. in [6]) as well as improve the infrastructure of the city by communicating these with the vehicles (Malik et al. in [7]). On the other hand, for the maritime case, the convergence of MTS and IoT has led to the promising IoT-improved MTS (IoT-MTS); however, the research is still under development and IT systems have not advanced so much to apply equally to ship cases like ground and air vehicles.

For example, Huang et al. in [8] propose a UAV-based system to smartly control and watch over real-time maritime areas in terms of complicated geographical, meteorological, and electromagnetic conditions. This control system offers functions such as automatic detection of ships violating rules, water area intelligent monitoring, water transportation order control, and water area pollution monitoring. However, the solution is exclusively for the safety of the ship, not for the cargo it carries.

Large unattended maritime areas are conducive to risks corresponding to piracy, interference attacks, or ransomware attacks which require ensuring the reliability and efficiency of information transmission in maritime areas. Therefore, one solution is to include blockchain technology that guarantees security during the journey. Yang et al. in [9] propose the keys to including this technology and operating in edge computing in an intelligent maritime transport system. The main keys in Yang et al. [9] are an IoT-enabled maritime transport communication system, which is a distributed system composed of base stations and offshore buoys and uses the unique structure of the blockchain to solve the problems of security and reliability in the network considering only the ships. While Zhang et al. in [10] propose an IoT-based collaborative processing system that unifies the modular structure and integrates multiple modules involved in modern maritime transportation systems. The proposal uses blockchain technology for transportation flow scheduling and management.

Despite advances in applications between IoT and maritime environments to improve safety on board, the trajectory of ships can have highly negative impacts on the management of IoT-MTS. Therefore, Hu et al. in [11] propose a transfer-learning-based trajectory anomaly detection strategy. However, we highlight that this detection is a posteriori; thus, any real-time correction can be made for the maritime route. The limited maneuverability in the trajectory anomaly detection, combined with the lack of efficient communication among ships and ports, maintain the risk of accidents and unwanted situations. In order to take action on the transport routes, we must have the position of the ships controlled at all times. The Global Navigation Satellite Systems (GNSS) system is currently the most attractive positioning system. The keys and challenges to using the GNSS system with autonomous vehicles within IT are presented in Jing et al. [12]. However, as mentioned at the beginning, this has been developed for the ground vehicle but is still in development for the maritime sector. In addition, for monitoring purposes, positioning and navigation are critical functions to determine the absolute and relative positions in the environment in which ships operate.

On the other hand, much interest has arisen worldwide in SatCom as the only means to globally cover the entire land and sea surface. Many operators and manufacturers have gradually built Low Earth Orbit (LEO) satellite constellations. Thanks to their characteristics of small delay and large coverage using constellations, this type of satellite provides a possible solution for the full coverage of the ground communication network. Traditionally, the satellite has also been preferred to provide maritime coverage. Instead, it was not until a few years ago that it gained greater importance thanks to the union with IoT. The work presented by Jin et al. in [13] provides a solution for the implementation of an IoT network supported by LEO satellite constellations, opening the way to the architecture of the IoT-SatCom. In addition, it provides solutions and research directions for the in-depth integration of the satellite-based devices.

We need to effectively allocate relatively limited onboard resources for meeting massive access demand when the satellite is integrated into IoT networks. Therefore, a global traffic model needs to be considered. The work performed by Qu et al. in [14] analyzes the characteristics of satellite-based IoT devices using geographic properties and applications where the design leads to the global traffic model of an LEO-based IoT network. This model has to be extended for maritime traffic.

In addition, from point of view of user traffic, the transmission of huge amounts of data generated from various sensors is the basis and prerequisite for the analysis and prediction of the marine ecosystem. The special geographical conditions of the ocean determine that its data transmission is different from the common terrestrial data transmission. Zong et al. in [15] present a study about marine environmental monitoring network architecture in order to collect environmental data from marine ecosystems using a heterogeneous network constructed by a multihop network and a satellite network. Considering the characteristics of satellite networks, such as high bandwidth-delay product, long latency, and high bit error rate, Zong et al. [15] proposes a transmission scheme that improves the transmission performance of Internet networks over satellite networks based on Transmission Control Protocol/Internet Protocol. Hence, providing an useful reference for the network model and data transmission for ocean monitoring.

Considering the global demands on IoT, the work presented by Yan et al. in [16] proposes a design for a satellite-terrestrial integrated architecture for the IoT relying on the software-defined network. Moreover, based on this architecture, they further propose a dynamic channel resource allocation algorithm to control the access of the IoT terminals with different priorities.

### 1.3. Our Contribution

We can see how the integration of the satellite with IoT networks is recommended for the transport sector, specially for maritime transport it is a potential candidate. Instead, there are several challenges that need to be addressed to include this architecture. The IoT-satellite union has been proposed for solutions on ships and their surrounding environments, however, it has not been proposed in detail for the cargo carried by ships. There are solutions for anomalies caused by external factors, but not in the cargo itself that the trajectories are not valid. The lack of maneuverability in the event of any anomaly or unforeseen event would cause the loss of the ship’s cargo. In addition, the inefficiency of communications with the ports prevents real-time route updates, causing delays in vessel departures and arrivals.

Against this background, we present two types of maritime transport services: monitoring the cold chain inside the containers and integrating data in Global Positioning System (GPS) and Automatic Identification System (AIS) protocols to improve communications with the ground segment through a MEO mega-constellation. A communications protocol and a proprietary messaging system based on IoT are defined for these services. Thanks to the proposed satellite network, the data are sent to the control center on land to order the update of the route of the ship in case of any problem in the load.

For SatCom’s estimation, we evaluated the communication parameters supported by a constellation of satellites in the MEO orbit that seeks to maintain the balance between delay and depot cost while guaranteeing 100% connection availability.

This proposal is framed on the one hand in the MEGALEO Project funded by Luxembourg National Research Fund (FNR) together with GOMSPACE as a partner. On the other hand, the 5G-Space Lab Project funded by the Luxembourg Space Agency (LSA) and the Ministry of Economy (MECO) of the Government of the Grand Duchy of Luxembourg.

The research hypothesis of “MegaLEO” is that a large (MEO and LEO) satellite constellation can operate semiautonomously by deciding and executing satellite and network operation configurations in space. In this direction, MEGALEO will focus on use cases where distributed control of the satellites’ orbital positions and radio resources would be imperative, matching the heterogeneous traffic demand across the globe (maritime in this case). 5G-Space Lab has provided the software for testing the constellation design.

The rest of the paper is organized as follows. The system model is presented in Section 2, the architecture for a satellite-based IoT system proposed for maritime transportation communications is presented in Section 3. The proposed services are defined in Section 4. The network dimensioning on the platform on the ship is shown in Section 5. The communication protocol are proposed in Section 6. Finally, we conclude in Section 7.

## 2. System Model: Maritime Communications and Internet-of-Ship

Communications for the maritime sector have historically used radio waves for the exclusive transmission of voice and data. Its main objective from the beginning has been to communicate distress and safety signals between coastal stations and ships. Secondly, with the advancement of telephony, they began to be able to communicate by voice with other vessels and receive weather forecasts by text. Subsequently, maritime radio communications experienced a great boom with the incorporation of satellites. With these systems there are radio beacons with which disasters or distress calls are located, likewise, there are satellite network systems that allow the use of telephony and data. Inmarsat is the operator that has traditionally offered maritime communications services via satellite. In addition, the satellite brought with it navigation systems, introducing systems such as Global Positioning System (GPS) and Automatic Identification System (AIS). In the context of this work, we are interested in the AIS protocol through the SatCom link. The AIS system is used to identify ships, their speed, and course at a specific time. This system, in maritime navigation, is a considerable element in the safety of ships, since throughout the navigation it makes the vessel visible, in front of large merchant ships. It also allows you to identify other ships that have AIS systems and see their course. Today, the AIS system is one of the most important systems in maritime communications since the radar was introduced. This system was designed with the intention that ships can be seen in any weather condition and thus avoid collisions and is widely used in the merchant marine. Two kinds of AIS systems are available:**Class A**: It is a more complex and quite expensive system that has different very high-frequency transmitters and receivers that are usually used by larger ships.-Frequencies: 156.025 MHz–162.025 MHz-Sending information: Continuous-Emission power: 12.5 W-Range: 50 Nm**Class B**: It is a lighter and more accessible system in terms of price. This class B AIS is the one used in recreational navigation.-Frequencies: 161.500 MHz–162.025 MHz-Sending of information: Every 3 min-Emission power: 2 W

The recent emergence of IoT technologies in mission-critical applications in the maritime industry has led to the introduction of the Internet-of-Ships (IoS) concept. IoS is a novel application domain of IoT that refers to the network of smart interconnected maritime objects, which can be any physical device or infrastructure associated with a ship, a port, or the transportation itself, with the goal of significantly boosting the shipping industry toward improved safety, efficiency, and environmental sustainability. Aslam et al. in [17] propose the architecture, key elements, and main characteristics for IoS in modern maritime communications era. The Aslam et al’s. work includes a literature review for IoS. At the present time, there are new digitalization and data exchange initiatives, providing new opportunities by better connecting operators, ports, people, and infrastructure through technologies such as IoS.

## 3. Proposed Architecture

The architecture to provide services in the transport of refrigerated containers through SatCom is designed and configured depending on the characteristics of the services provided on the ship. Hence, it is composed of three wireless networks with different technologies as shown in the Figure 1. These subsystems or subnetworks are:**WSN (Wireless Sensor Network):***On-board cargo monitoring subsystem*. It is a network of sensors inside container ships based on Zigbee as a representative of the IoT family protocols. It is responsible for providing the data collected from the monitoring of the refrigerated container sensors.**WLAN (Wireless Local Area Network):***On-ship communications subsystem*. It is responsible for supporting communications on the ship (via WIFI), without being affected by corrosion of the wiring due to weather conditions. They are cheaper than wiring container ships as well as being more adaptable to the environment (mobility on the ship).**WWAN (Wireless Wide Area Network):***Global communications subsystem*. It is provided by a constellation of MEO satellites to communicate the container ships with the maritime control center on the ground segment. This link uses the AIS protocol over satellite.

### 3.1. IoT (Zigbee) Based WSN Subsystem

In the new IoS paradigm, sensor networks allow the monitoring of different parameters of refrigerated containers in maritime logistics, such as temperature and relative humidity, the object of the solution proposed in this work. To this end, the proposed WSN consists of a wireless sensor-based network that would comply with ISO 10368:2006 standard, which establishes a series of requirements regarding communication between devices, data records, and other standards. The proposed technology to be able to communicate these sensors will be through Zigbee due to:1.The Zigbee standard operates in license-free bands that facilitate the integration of services in the transport sector in the 2.4 GHz, 900 MHz, and 868 MHz bands. It offers speeds of up to 250 kbps and 65,000 connected devices (enough to monitor the temperature and relative humidity of all containers on the ship).2.Zigbee’s ability to support mesh networks that, due to their decentralized nature, are ideal for the structure of containers carried by ships. In addition, each node is capable of self-discovery on the network. When nodes leave the network, the mesh topology (not available in Bluetooth) allows nodes to reconfigure routing paths based on the new network structure. Mesh topology and ad-hoc routing features provide greater stability in changing conditions or in the event of a single node failure. This feature can be beneficial in transportation and container logistics since it would frequently vary from place to place when used in intermodal transport.3.Lower power consumption compared to that required by Bluetooth. When it is transmitting, Zigbee offers consumption of 30 mA and at the rest of the time just 3 μA. On the other hand, Bluetooth consumes about 40 mA transmitting and 0.2 mA at rest. This is because the Zigbee system spends most of its time in a “sleep” state, while Bluetooth is always in a transmit and/or receive state. In this way, it is optimal in terms of energy savings and battery life, something necessary in long-term transoceanic trips.

For WSN, according to the Zigbee standard defined in IEEE Standard [18], we find three types of devices or nodes as shown in Figure 2.

These nodes are:Coordinator:-There can only be one per network.-Starts the formation of the network.Router:-It is associated with the network coordinator or with another ZigBee router.-It may act as coordinator.-It is in charge of multihop routing of messages.End device:-Basic element of the network.-It does not perform routing tasks.

In order to calculate the distribution of the IoT sensors on the ship, it is necessary to have the stowage arrangement (placement of the cargo) in a container ship. An example of container layout is shown in Figure 3. It is common to separate cargo both on deck or ship’s hold, into different stacks (or blocks) which are differentiated into bays, rows, and tiers according to nomenclature defined by D. Steenken et al. in [19]. Each stack can contain both normal containers and “refer” containers (referred to as refrigerated containers). The measures of the standard TEU container (twenty-foot equivalent unit) are 20 feet, corresponding to 6 m long, 2 m wide, and 2.5 m high as shown in Figure 3. Based on these measurements, 200 containers are grouped on deck in each stack, storing 4 tiers high, 10 rows wide, and 5 bays deep. The layout of the containers is observed from a frontal perspective (10 rows and 4 tiers of containers). The side view forms a cube of containers (5 bays of containers). In this way, the layout of a container-type stack (on deck) and the corresponding measurements are displayed.

In summary, the container type stack on deck is made up of 10 rows of containers placed horizontally, 4 tiers of containers placed vertically, and 5 bays of containers placed deeply.

In the ship’s hold, the design changes as half of the containers can be accommodated by each stack or block, since the height is reduced by half, each stack can hold about 100 containers. In total, the design of the stacks placed along the ship can accommodate 1200 containers. In order to generate a Zigbee network, the previously shown grouping type will be taken as the type network. In no case, the distance of 75–100 m (allowed by the Zigbee network) between end nodes and coordinator node is exceeded.

In this case, the design of the WSN architecture requires the following conditions:Being able to accommodate a maximum of 200 reefer containers in each Zigbee network.A decentralized mesh network is proposed so that each node is capable of self-discovery in the network (this will facilitate the transit of monitored reefer containers). This topology allows the nodes to reconfigure the routing routes each time new ones are included or they disappear (in the case of transit or failure of a reefer).A Zigbee coordinator node is included for each cluster with a maximum of 100 (ship´s hold)/200 (deck) containers, in charge of forming the Zigbee network. It establishes a communications channel and a network identifier (PAN ID). Handles packet routing of the final nodes. In addition, the same coordinator node will serve as a gateway in the next WLAN subsystem.A Zigbee end node is implemented for each refrigerated container in each cluster.There is a sensor capable of measuring the relative humidity and the temperature of the container.

The main characteristics taken from the Zigbee protocol in IEEE Standard [18] for our proposal of the WSN subsystem are listed in Table 1.

### 3.2. Wi-Fi-Based WLAN Subsystem

The second subsystem, WLAN, is shown in Figure 4. This subnetwork consists of WIFI5 technology according to IEEE IEEE Standard [20], whose elements are detailed as follows:Gateways or coordinating nodes (in the WSN network) for each group of containers (they share the 2.4 GHz frequency of the Zigbee-based WSN network as the aforementioned 5 GHz for WIFI5). These devices will be redundant.Automatic Identification System and geolocation antenna (AIS + GPS transponder) class A based on WIFI5 technology with a redundant access point (AP). Commercial solution found: Ray-Marine with the AIS4000 product, which is a class A AIS transponder with built-in WiFi and GPS.Control system based on PDAs for ship crew with WIFI5 in which to monitor the previously described WSN sensor network.Concentrator router (redundant) of the rest of the previously described elements located in the command bridge.The ship’s electrical power line serves to provide sufficient energy to all the elements of the network. There is a redundant element through a UPS (uninterruptible power supply) to guarantee the power supply.

The main characteristics of WiFi for the WLAN subsystem taken from IEEE Standard [20] are listed in Table 2.

### 3.3. Satellite Based WWAN Subsystem

A communication link based on a MEO satellites constellation has been chosen to provide coverage on the ship itself, as well as the ship with any data monitoring point on land, in order to find the balance between delay and cost of deployment for use in this WWAN (Wide Wireless Area Network).

As shown in Figure 5:The ship has a satellite antenna that allows both transmitting and receiving in the frequency bands used by this satellite, which will be connected to the communications rack established in the wheelhouse.Shore-based satellite base stations that function as gateways for ship-to-shore-to-ship communications.These base stations are connected to the backbone network to provide an IP connection.Once a connection to the data backbone is provided, data is sent to a cloud server to secure the data transmitted by the ship (WSN data monitoring, AIS and ship geolocation, and any other necessary ship communications -Voice and Data-).An application is provided to visualize the data received by the ship.

To find a balance between satellite link delay and the cost required for system deployment, a medium earth orbit (MEO) satellite constellation designed to provide low-latency broadband connectivity to remote locations for mobile network operators and maritime service providers is chosen.

A total of 20 satellites are deployed in a circular orbit along the equator at an altitude of 8063 km at a speed of 11,755 mph (18,918 km/h), similar to that used by SES with O3b in [21,22]. The deployed constellation is shown in Figure 6.

Each satellite is equipped with fully steerable Ka-band antennas at 19.7 GHz for the downlink and 24 GHz for the uplink. Round-trip latency is 140 ms for data services.

The shipping route crosses the Atlantic from Southern Europe to the Caribbean area. Regarding the service area, as shown in Figure 7, the sea route is always within the standard service area, thus ensuring its connectivity.

## 4. Maritime Transportation Services

The evolution of IoT in the field of maritime transport (“Internet of Ships”) has gained special importance, as argued in Section 1, and this has generated new areas of application in said industry. In this context, this work presents two new services supported by IoT and a satellite network for logistics transport at sea that has benefited society.

### 4.1. Monitoring Service of Transported Cargo

This service consists of creating a communications network based on IoT and a constellation of MEO satellites to monitor the cold levels of cargo transported by containers on ships to guarantee its conservation by maintaining the cold chain and optimal relative humidity conditions (necessary in the transportation of medicines and perishable food).

Figure 8 presents a scheme for the proposed service. In this case, a thermometer sensor and a hygrometer capable of monitoring the temperature and relative humidity of the load are detailed. In addition, as can be seen, this scheme is included within the wireless sensor network WSN, which will be detailed in the following chapter:

This service must take into account the following:Have sensors capable of measuring the following parameters:-Thermometer sensor, capable of measuring temperature ranges between −25.0 °C and +25.0 °C (with a precision of 1 decimal).-Hygrometer sensor, capable of measuring relative humidity ranges between 0% absolute dry air and 100% air completely saturated with water vapor.Regulations to be met by the sensors placed according to the Agreement on the International Carriage of Perishable Food Products (ATP) (UNCTAD) [17],-Standard UNE EN 13846—Temperature recorders and thermometers for the transport, storage, and distribution of refrigerated, frozen, and deep-frozen foods, and ice cream. Periodic verification-.-Standard UNE EN 12830—Temperature recorders for the transport, storage, and distribution of temperature-sensitive products. Tests, operation, aptitude for use-.

### 4.2. Enhanced Positioning Service

The power lines installed on container ships (Power Line Communications, PLC) suffer from deterioration caused by corrosion on the high seas, by handling the loading/unloading of containers in ports (stowage), and have limitations both in speeds of transmission, as well as how little immune they are to noise. In this way, a wireless communications service capable of covering both the monitoring service through sensors and other services that this type of transport can use (such as GPS and AIS) is offered. For this, the solution must take into account the regulations that must be complied with in wireless communications on the ship that define the exchange of information:ISO 10368. The main summary lists the interfaces and protocols necessary to achieve this certification.Standardized architecture, correct dimensioning, and messaging protocol that defines correct monitoring of the cargo transported, as well as the communication of other services on the ship.

Therefore, the solution to respond to this service is based on designing and developing the technological evolution that responds to the aforementioned problems generated by wired communications.

## 5. Network Dimensioning on the Platform on the Ship

The necessary dimensioning within the ship’s communications is shown in Figure 9. On the one hand, how the available spectrum is distributed both for the WSN subsystem and for the WLAN subsystem is considered, and what channels will be used to transmit both from the endpoints located in the containers and the channel used by other possible services such as AIS, and GPS. The data are centralized in a central node (communications rack) which groups all the information before being sent to the satellite.

### 5.1. WSN Subsystem

For the wireless sensor area network WSN is dedicated to monitoring through Zigbee, a different communication channel assuming Frequency Division Multiplexing (FDM) is used for each type of container grouping (200 on deck and 100 in the ship’s hold). In this way, the IEEE 802.15.4 standard according to IEEE Standard in [18] that allows users in the 2.4 GHz band is distributed as follows in this solution:Number of channels available: 16 channels separated in their central frequency ($f_{c}$) by 5 MHz (80 MHz of available spectrum). Thus, each coordinating node is assigned one of 16 channels. The 16 channels are never exceeded since there is a limitation of space on the ship for each group of containers.Frequency bandwidth of about 3 MHz for each channel.Each node has a spectrum of 1.23 kHz. According to the standard in the total 80 MHz spectrum, 65,000 endpoints can be arranged.A maximum number of endpoints per channel: 4065 nodes (not exceeded since each group type allows a maximum of 200 containers).

Here is how the Zigbee spectrum is used to cover clusters of containers along the ship with different channels: Example of channels for 1200 containers placed in stacks:1.Zigbee channels from 1 to 4 for 4 stacks of 200 containers on deck (800 containers in total).Channel 1: $f_{c}$ = 2405 MHz.Channel 2: $f_{c}$ = 2410 MHz.Channel 3: $f_{c}$ = 2415 MHz.Channel 4: $f_{c}$ = 2420 MHz.2.Zigbee channels 5 to 8 for 4 stacks of 100 containers in the warehouse (400 containers in total).Channel 5: $f_{c}$ = 2425 MHz.Channel 6: $f_{c}$ = 2430 MHz.Channel 7: $f_{c}$ = 2435 MHz.Channel 8: $f_{c}$ = 2440 MHz.

The protocol used is that established by the Zigbee standard, where each end node (the one used for monitoring each reefer container) transmits on its assigned channel. To identify each node, the license plate (inscribed on the container door) which is marked on each reefer and its spatial reference is used, so that the onboard crew can configure it in each stowage both in its Channel (CH), and assign it an identifier (reefer registration) at the application level (see explanation later in the chapter of communication protocols).

### 5.2. WiFi Subsystem

On the other hand, for the WLAN subsystem, technology has been established that allows communication within the area of the container ship itself that mainly does not interfere in the Zigbee band (2.4 GHz) and that provides sufficient coverage for each stack of containers designed (let us remember 200 containers on deck and 100 in the hold) and other multiplexed services such as AIS and GPS.

In this way, the choice of the WIFI5 standard (802.11ac), allows us to solve the problem of interference since by operating in the 5 GHz band the signals emitted do not interfere with those of the end nodes installed in each reefer.

Therefore, by broadcasting in the 5 GHz frequency, the gateways that are going to be used (remember that they will also be used as coordinating nodes in the Zigbee network), have new communication channels and available spectrum. When using FDM, this spectrum will be distributed in bands of 80 MHz per channel (remember that more spectrum can be dedicated per channel thanks to the use of this standard), in such a way that, of the 160 MHz possible usable, the spectrum will be divided into two channels of 80 MHz each. Thus:1.A channel is guaranteed to collect information from each of the channels provided in the Zigbee network (WSN). Since 80 MHz is available, 40 MHz is used in the design example provided for the WSN communication protocol. If new reefer-type pools are added to the container ship, more spectrum is still available to collect the new monitoring channels.2.Another 80 MHz spectrum is guaranteed on a different channel for communications within the ship that collects information received from both the AIS and the GPS coordinates of the ship.

## 6. Communication Protocols

In order to establish a communication protocol, we start from the base of the proposed architecture referring to the ship (WSN and WLAN)—it will be detailed from the smallest to the largest wireless network. Access to the medium will be based on CSMA/CA (Carrier Sense Multiple Access with Collision Avoidance) in both and will be defined in each of the wireless networks. Lastly, a message protocol will be established to define the communication between services (monitoring, identification, and position of the container on the ship and information related to AIS and GPS).

### 6.1. Channel Reservation Protocol

The protocol is based on the IEEE 802.15.4 standard [18], which defines CSMA/CA (Carrier Sense Multiple Access with Collision Avoidance) as the Media Access method for each grouping type designed (200 on the deck of the ship or 100 containers in the ship’s hold in each case). It is a random waiting mechanism that avoids collisions in the transmission of each Zigbee node.

Detailing step by step shown in Figure 10, the following flow is followed:1.Each container pool broadcasts on a different channel.2.Within each channel, the end nodes that monitor each reefer listen to the medium.3.If the medium is free, they reserve the channel to request connection data from their coordinator, if not, they wait a random time and try again.4.If they have data to transmit and have not yet reserved the channel, they listen to the medium again and when it is free they ask for a temporary space to be able to reserve it.5.If the end node already has the reserved channel and has data to transmit, it waits for a random time and then starts the data transfer.

Below, the association procedure of the sensor with its coordinating node is detailed, as well as the following procedure that will be chosen for sending data.

#### 6.1.1. Sensor Association Procedure with Its Coordinator

In the following Figure 11 it is observed how before sending data, within the 802.15.4 standard there is the association procedure in which the final node (in our case the Zigbee device placed in the reefer), requests the data of the PAN ID (Personal Area Network Identification) to the coordinating node. Later, it will be seen how once associated, the sending of data from the final node to the coordinator will continue and how this communication is carried out.

The CSMA/CA method used for the WSN network uses control frames to able to coordinate the sending of data on the channel, where:**SIFS**: Short Inter Frame Space. The time interval between transmission of packets (mandatory period of idle time on the transmission medium).**DIFS**: Distributed Inter Frame Space. Minimum delay for asynchronous traffic competing for access. An end node waits for a DIFS before transmitting any monitored data when the channel is free.**RTS**: Request to Send. Control packet for data transmission request.**CTS**: Clear to Send. Control packet indicating that the receiving node is free to send.

In this way, by using the control frames explained previously, it is established a procedure -after association- able to send data from end nodes to the coordinator in the WSN.

#### 6.1.2. Sensor Data Submission Procedure with Its Coordinator

As we have seen previously, Figure 11 details the procedure for associating an endpoint (final node) placed in the container with its coordinator node. Now, Figure 12 details the channel reservation process to send data (in this case those that come from monitoring, as well as those that identify the container). In addition, a designed stack of containers located on the deck is observed in detail. They use the same channel (in this case Channel 1), with a bandwidth of 3 MHz (separation between central frequencies of each channel of 5 MHz), and a reefer identifier assigned as ID (registration of the container) to be able to recognize it within the Zigbee mesh.

#### 6.1.3. Procedure for Sending Data in WLAN of the Services Provided

Once the channel has been reserved and the useful data of the sensor network has been transmitted to its coordinators, it is time to see how the channel reservation is made in the following wireless local area subsystem -WLAN-. WIFI5 is the technology chosen, using the previous subsystem detailed with the same access to the medium (CSMA/CA). Unlike the WSN network, it uses beaconing (virtual carrier) through a vector of reserve (NAV). In this way, all the gateways placed on the ship follow the next steps as shown in Figure 13:1.They listen if the medium is free and if it is not, they wait. The gateways update their reservation vector of the NAV network (it works as a countdown, from which moment you can try to send data again).2.Once the medium is free, they wait for an IFS time (Inter Frame Space, or frame time).3.After this IFS time, they listen to the medium again, and if they find it still busy, they wait for another IFS time.4.If, on the other hand, when listening to the medium after the IFS time, it is free and its reservation vector of the NAV network is 0, the sending of data begins.5.Once the data is sent, the end node waits for a data confirmation ACK received by the concentrator router of all ship communications. Buffer is released.6.If this ACK is not received, the data frame is forwarded.

The gateways (coordinating nodes in the sensor network) placed throughout the ship, encapsulate the received frames (both from the sensor network and AIS and GPS). In the MAC layer frame headers in the IEEE 802.11 standard, a duration field is marked that specifies the transmission time required for the frame, in short, the time that the medium will be busy. Gateways in the ship’s WLAN mesh listening on the wireless medium read the duration field and set its NAV (Network Allocation Vector), which is an indicator to a gateway of how long to defer access to the medium.

A WLAN network gateway (WIFI5) listens to the free medium, and then sends data as shown in Figure 14. Meanwhile, the rest of the Gateways wait for the channel to be free and after a random time, they send the received data (sensors/AIS/GPS). It is also observed, how the data sent from the sensors is received by different gateways (mesh network) with the destination address of the router, and how the AIS/GPS data is sent directly to the same router, before being sent to the satellite (next WWAN subsystem).

### 6.2. Messaging Protocol

For the architecture proposed, also we define a message protocol within the ship. This messaging protocol will allow us to:1.Identify the data monitored by the sensor related to temperature and relative humidity.2.Locate the position of the monitored container inside the ship, so that action can be taken quickly in the event of failures.3.Identify with a reefer container ID that allows monitoring (in a cloud application) the sensor data (humidity and temperature).4.Identify what data comes from each different service.

A stack of containers is designed as shown in Figure 15, in which 200 containers are arranged on the ship´s deck using the spatial form of a cube. Figure 15 shows the placement and spatial form definition of the container stack:10 rows of containers placed horizontally named “R”.4 tiers of containers placed vertically named “T”.5 bays of containers placed deeply named “B”.

At the time of stowage (loading and connection of the reefer to the power line), the manual configuration at the application level of the Zigbee sensor placed in the container is carried out. To do this, in the different layers of the 802.15.4 standard, there is a payload and different headers (control and addressing) that are added to each defined layer, as can be seen in Figure 16:

The payload at the Application Sublayer level (ASDU -Application Service Data Unit-) has a variable capacity depending on the use or not of headers in other layers. Through the application and the monitoring of the sensor itself, a minimum space will be reserved so that the information shown in Figure 17 can be transmitted:


**1. Information received from the sensor referring to its humidity and temperature (SENS):**


A minimum of 12 octets (bytes) reserved in the ASDU payload, will be given by the information gathered from the sensors placed in the monitored reefer container. Although only 3 bytes in binary encoding are used for monitoring, a further 9 bytes are reserved for future use.

Thermometer sensor (2 bytes):-1 byte: integer part of the temperature (range −25 °C, +25 °C). The most significant bit of the 8 bits will tell us if the temperature is negative or positive:∗1: positive value.∗0: negative value.-1 byte: decimal part of the temperature.Hygrometer sensor (1 byte):-1 byte: the relative humidity (number from 0–100).

Example:Temperature: −23.7 °C-1 byte: (-): 0 –> 23: 0011001-1 byte: 7: 00000111Relative humidity: 85-1 byte: 01010101

Figure 18 below shows the coding of the 3 bytes used in this example of monitoring the temperature and relative humidity of the container:


**2. Information referring to the identification of the monitored reefer container (ID):**


As seen before, the container license plate is entered in the application when the storage is carried out and is encoded in ASCII code (alphanumeric characters), devoting a maximum of 10 bytes (the plates that identify the container do not occupy more than 10 alphanumeric characters).

Container license plate example: MAE 555-321

MAE: referring to the shipping company that owns the container (Maersk).555-321: identification of the container.

In this way, Figure 19 shows the detail of the information referring to the identification of the container (ID) of the example exposed with the corresponding ASCII encoding:


**3. Information referring to the spatial position of the monitored reefer container within a container type grouping:**


To identify the position of the container within the different stacks of containers along the ship’s deck/hold, 4 binary coded bytes are dedicated to helping an operator to locate it. For it:1 byte: location of the stack of containers in order to identify it on the ship (up to 255 possible locations).1 byte: row number of the stack of containers (R).1 byte: bay number of the stack of containers (B).1 byte: tier number of the stack of containers (T).

An example of the spatial location of the container is shown in Figure 20, and the following Figure 21 shows how the message is manually encoded in the ASDU during stowage to choose its position within its grouping.

In this example the following information travels:Number of a stack of containers location (1 byte): Stack 1: 00000001Row identification (R) (1 byte): Row 10: 00001010Bay identification (B) (1 byte): Column 1: 00000001Tier identification (T) (1 byte): Height 4: 00000100

Therefore, the container position message that will go in the ASDU payload will be as follows:


**4. Information referring to identifying the services in the communication:**


Once the communication protocol of each of the nodes that make up the WSN network has been defined, it will be explained below how this network will communicate with the WLAN network through WIFI5. It is established that, in each Gateway placed along the ship, depending on whether it receives information from the sensors or from the GPS or AIS service, this information is encapsulated and encoded according to the 802.11ac (WIFI5) standard, the first bits of the message data frame. In the case of encapsulating the sensor network data, it will be done by the specification of the WiFi standard such as Zigbee, adapting the network cards of the Gateways. Once encapsulated, an encoding is defined in the payload of the message at the application level. In this way, it will be possible to know what data from which services are being collected and issued, as illustrated in Figure 22:

### 6.3. Satellite Link

The satellite constellation has been tested using STK and MatLab software. We obtain different parameters such as visibility time and communication time between ship and satellite and carrier-to-noise ratio (CNR). With these results, we validate the requirements for services in the design.

Table 3 represents the main parameters of the satellite communications system that is used to provide round-the-clock coverage of the sea route.

Based on the MEO orbit constellation proposed in Section 3.3, Figure 23 depicts the 24-h access time to each of the constellation’s satellites from any point along the maritime route. It can be seen that with the current configuration, access to at least one satellite is always available throughout the 24 h, ensuring 100% availability.

On the other hand, Figure 24 represents the histogram of the carrier to noise ratio (C/N) obtained during the visibility time for Satellite 11. It is observed that in the downlink the C/N can vary from 2.4 to 5.7 dB, while in the uplink the range is from 0.4 to 4.2 dB.

## 7. Conclusions

In this work, an architecture based on three types of wireless networks has been proposed to solve several current problems in maritime transport, such as the transport of merchandise under special conditions (products under specific cold temperatures). The solution will be valid to provide a cold chain monitoring service through an IoT network supported by a constellation of MEO satellites.

The networks that make up the architecture are WSN, WLAN, and WWAN which use Zigbee, WiFi, and SatCom technologies respectively. In the proposed solution, the communication between the sensors within the WSN network has been detailed, forming an IoT network. In turn, the network of sensors communicates with the base station located on the deck of the ship via WiFi to subsequently communicate through the MEO satellite constellation with the control center on land. This communication with the ground segment is done through the classic maritime communications protocol, the AIS protocol. The proposed design presents the advantage in the throughput of the system by unifying existing protocols in maritime communications such as AIS with a new, more simplified protocol. In addition, we highlight the use of MEO satellites to reduce the latency that current systems based on GEO satellites produce.

On the other hand, the communications protocol and the messaging system have been defined to send the information from the sensors to the control center. This information within the service is the temperature and relative humidity of the containers. The messages defined in this solution are integrated into the AIS protocol messages to forward them through the satellite link. Unlike other works of the state of the art, the architecture proposed together with the protocol and communications messages allows updating the route of the ships so as not to lose the merchandise in the event that the cold chain is broken, updating the destination port.

The messaging system also offers the option of extending the messages in the future to include other physical parameters of the refrigerated containers, such as the levels of O_2_/CO_2_ in the container, and volumetric levels of the cargo transported. Therefore, Zigbee frames have space reserved for future expansion.

Future work will include further analysis and optimization of the satellite communications system, a more extensive study of LEO, MEO and GEO orbit compensation, and a more detailed study of rain attenuation.

## Figures and Tables

**Figure 1 sensors-22-06450-f001:**
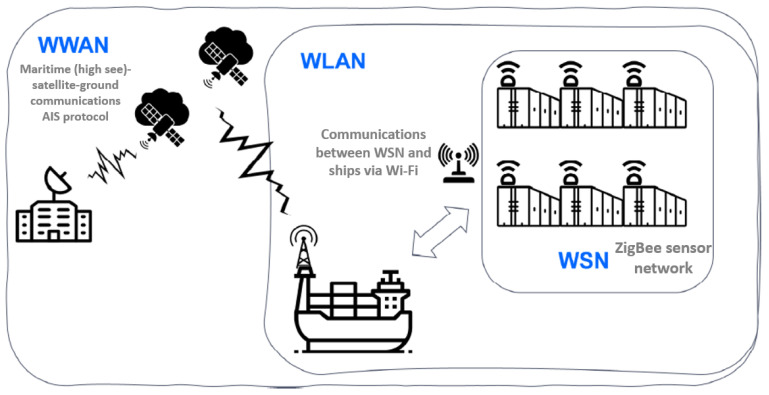
Proposed architecture scheme.

**Figure 2 sensors-22-06450-f002:**
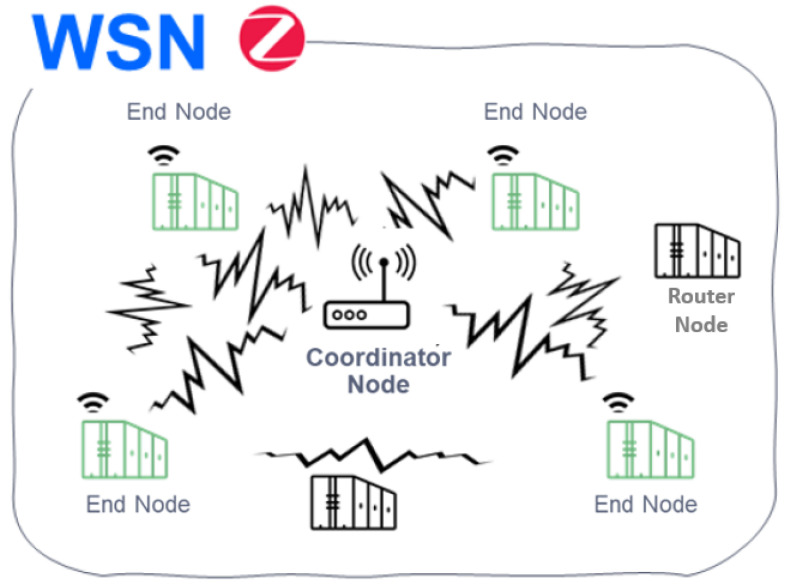
Distribution of node for the WSN subsystem.

**Figure 3 sensors-22-06450-f003:**
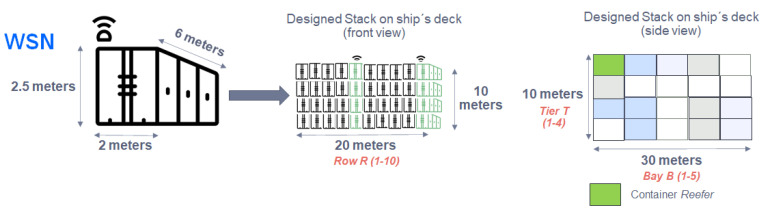
WSN subsystem scheme.

**Figure 4 sensors-22-06450-f004:**
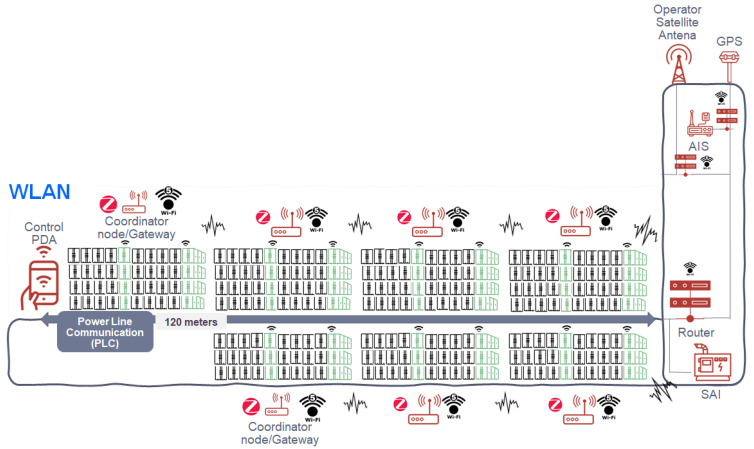
WLAN subsystem scheme.

**Figure 5 sensors-22-06450-f005:**
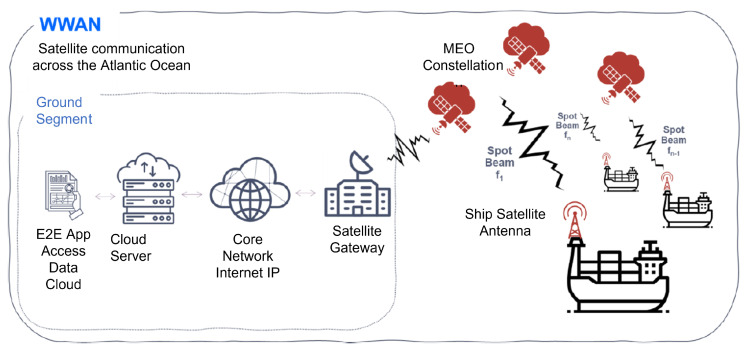
WWAN subsystem scheme.

**Figure 6 sensors-22-06450-f006:**
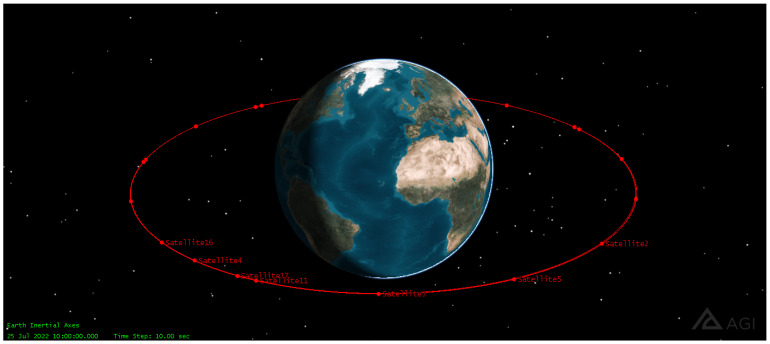
As part of the constellation, a total of 20 satellites are deployed in a circular orbit along the equator at an altitude of 8063 km in MEO orbit.

**Figure 7 sensors-22-06450-f007:**
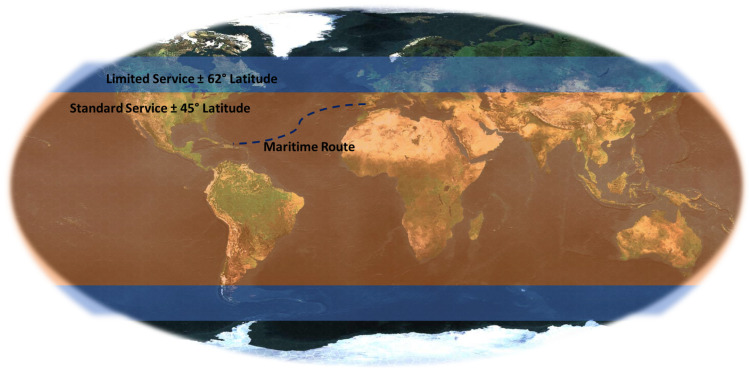
Maritime route and service area.

**Figure 8 sensors-22-06450-f008:**
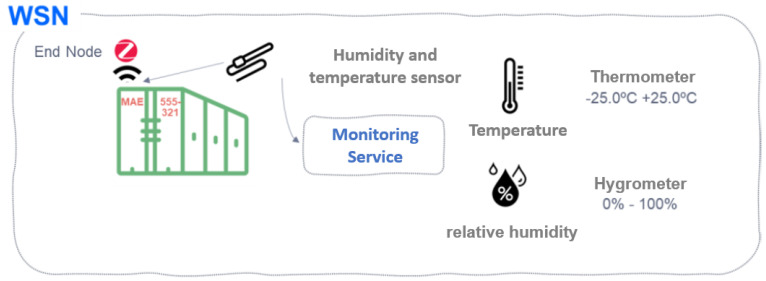
Monitoring of temperature and relative humidity in the container.

**Figure 9 sensors-22-06450-f009:**
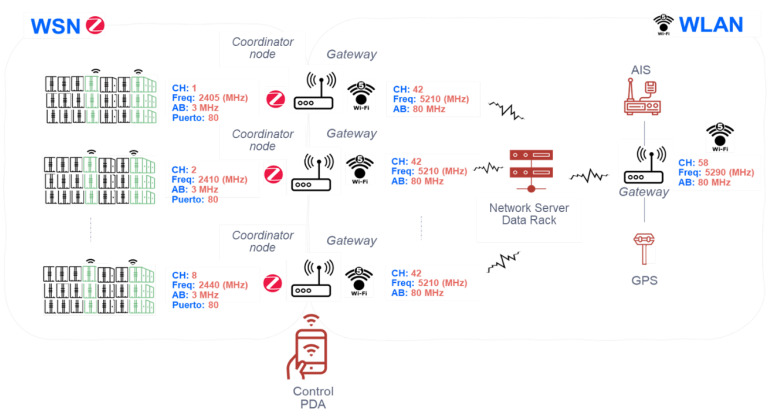
WSN/WLAN subsystems dimensioning across the ship.

**Figure 10 sensors-22-06450-f010:**
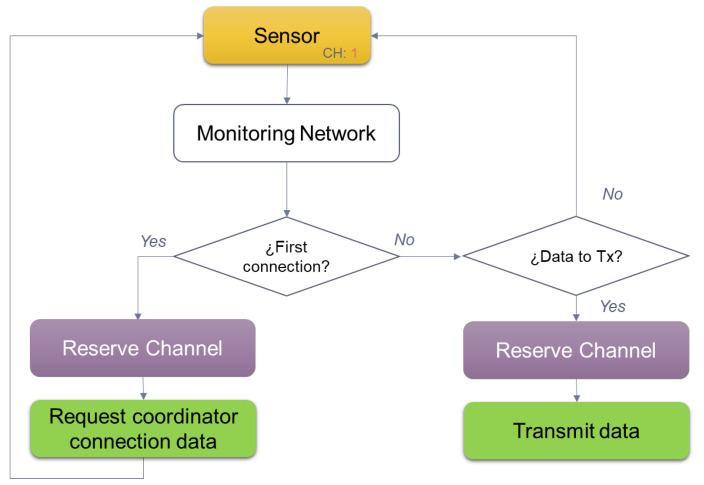
Sequence for the reservation of the medium by a sensor in the Zigbee network.

**Figure 11 sensors-22-06450-f011:**
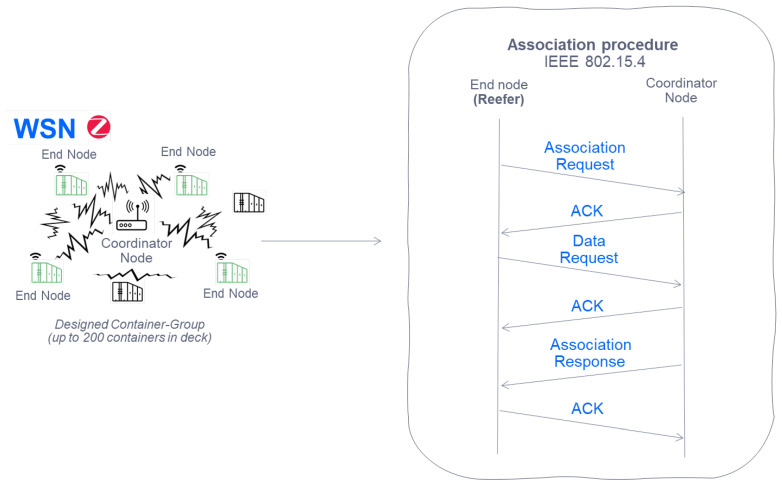
IEEE 802.15.4 Sensor association procedure with its coordinator node.

**Figure 12 sensors-22-06450-f012:**
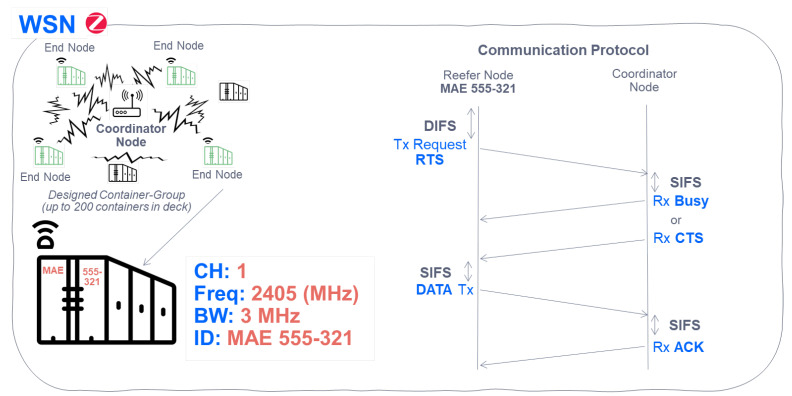
Access to the Medium for data transmission in the WSN Reefer-Coordinator network.

**Figure 13 sensors-22-06450-f013:**
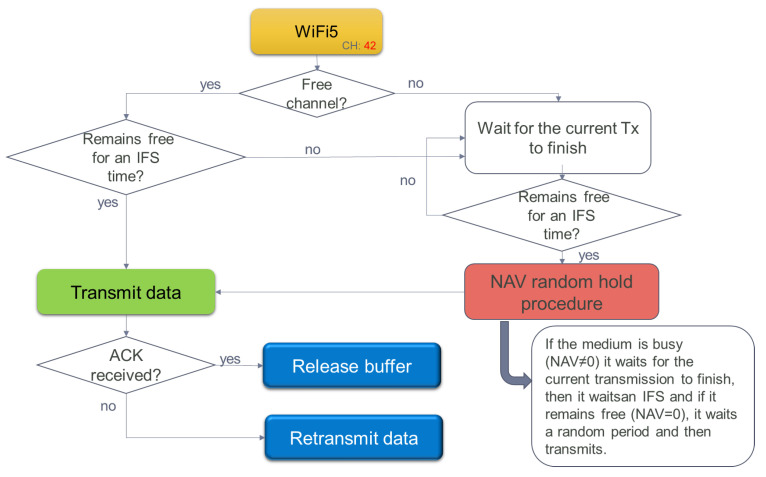
Distributed Coordination function in WLAN transmission gateways.

**Figure 14 sensors-22-06450-f014:**
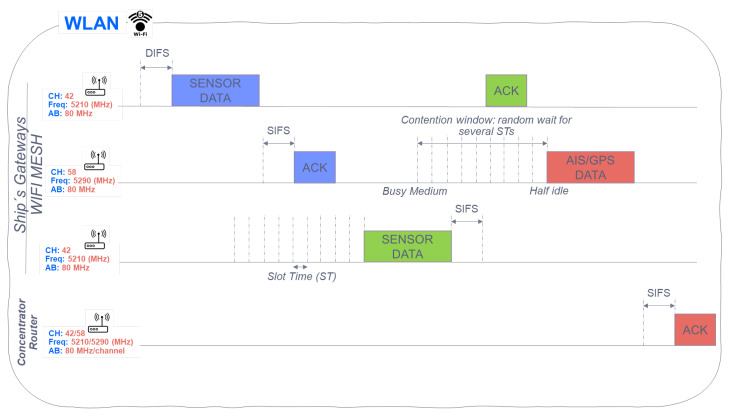
Sensor data sending detail, AIS, GPS in WLAN.

**Figure 15 sensors-22-06450-f015:**
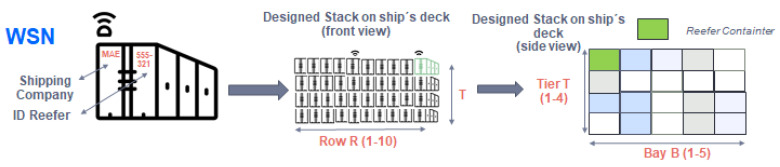
Design of a container stack with the spatial location of the monitored reefer container.

**Figure 16 sensors-22-06450-f016:**
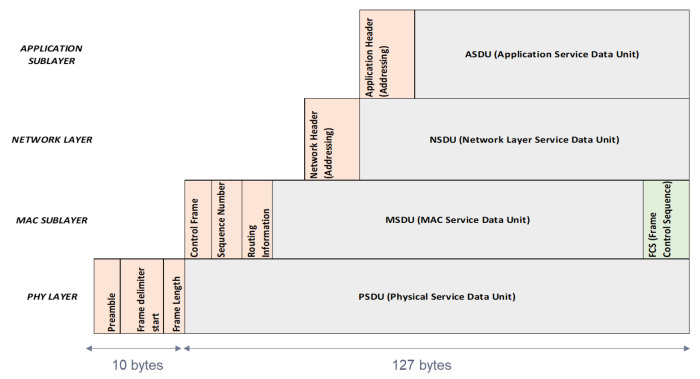
Data Units in the layer structure.

**Figure 17 sensors-22-06450-f017:**
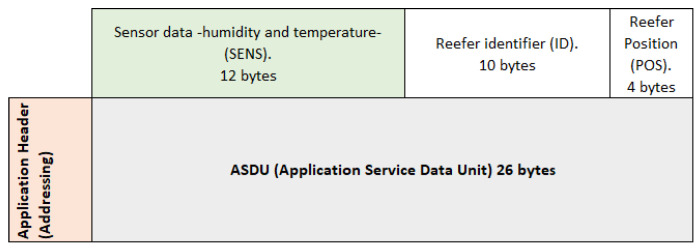
Zigbee Application protocol frame bit reservation for container monitoring.

**Figure 18 sensors-22-06450-f018:**
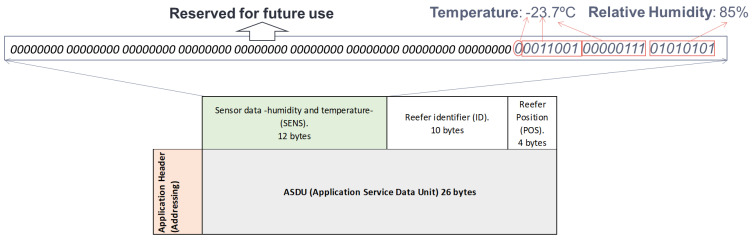
Detail of the message with the temperature and relative humidity of the container.

**Figure 19 sensors-22-06450-f019:**
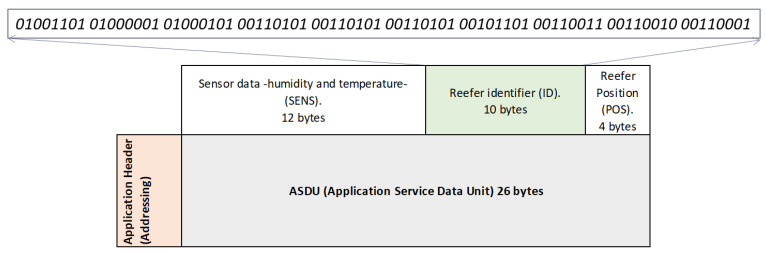
Container ID Message Detail.

**Figure 20 sensors-22-06450-f020:**
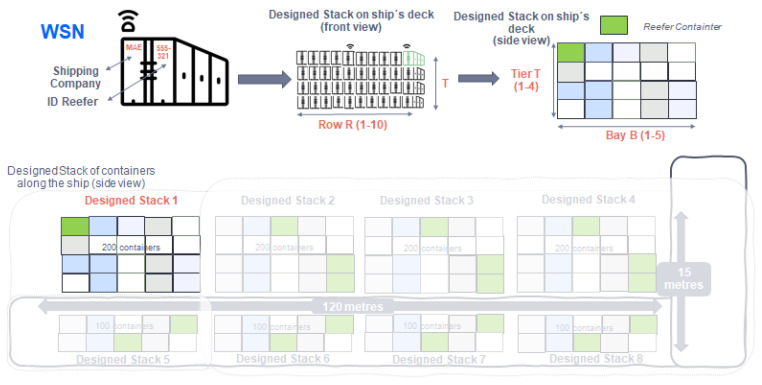
Manual configuration in the storage of the container location.

**Figure 21 sensors-22-06450-f021:**
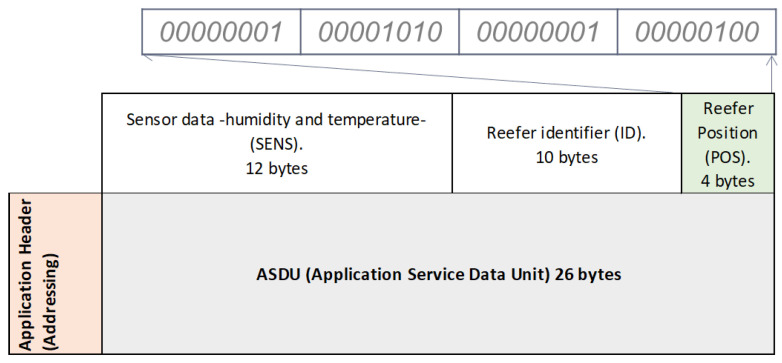
Detail of the message with the position of the container on the ship.

**Figure 22 sensors-22-06450-f022:**
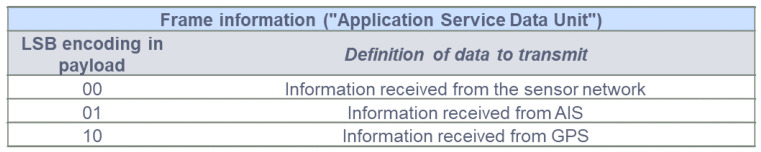
Coding in IEEE 802.11ac payload the services provided.

**Figure 23 sensors-22-06450-f023:**
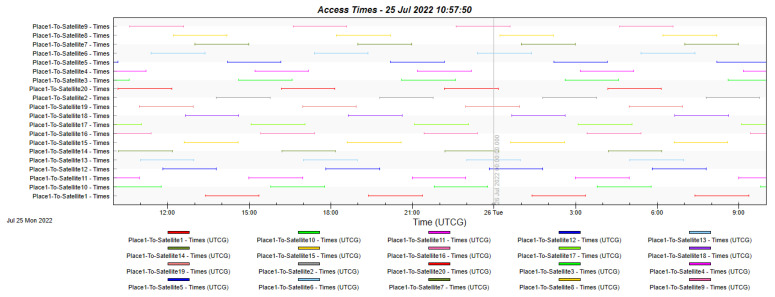
Availability of service at any point along the maritime route.

**Figure 24 sensors-22-06450-f024:**
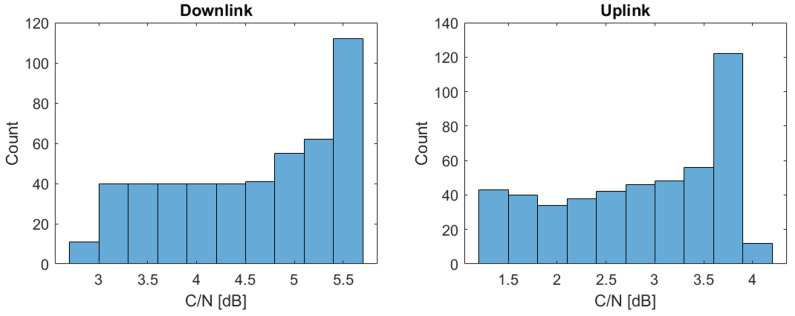
Carrier to Noise Ratio histogram in the maritime route.

**Table 1 sensors-22-06450-t001:** Zigbee parametrization.

IEEE Standard	Bit Rate	Latency	Frequency	Range	Number Dev	Topology	Battery Life	Security	Cost	Complexity
802.15.4	20–250 kbps	30 ms	2.4 GHz	10–75 m	2–65,000	Mesh Star	>1 year	128 bits AES	Low	Simple

**Table 2 sensors-22-06450-t002:** WIFI5 parametrization.

Standard	Release	Frequency	Bit Rate	Indoor Range	Outdoor Range	Bandwidth
WIFI5 802.11ac	14	5 GHz	3.5 Gbps	100 m	50 m	20, 40, 80 + 80, 160 MHz

**Table 3 sensors-22-06450-t003:** Communications Satellite System Parameters.

Orbit	MEO
Number of Satellites	20
Altitude	8063 km
Downlink Frequency	19.7 GHz
TX power	30 dBW
TX Gain	43.69 dB
Uplink Frequency	24 GHz
Latency	140 ms

## Data Availability

Not applicable.

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
