# Peer review of "Enhanced Communications on Satellite-Based IoT Systems to Support Maritime Transportation Services"

_sensors, 2022, doi:10.3390/s22176450_

Round 1

Reviewer 1 Report

Acronyms on the abstract are not a good choice. Please, define it or think of another way to introduce it in the abstract.

Keywords need to be reviewed. I suggest using until 5 and do priority for more auto-explained keywords than acronyms

The paper organization can be improved. I suggest splitting the introduction into 3 sections (Introduction, Background, and Related Work) 

The proposal is interesting, mainly in terms of practical ways. It is not clear the scientific contribution, but the environment is well-defined and explained, helping other researchers to be deep in terms of protocols, architectures, and technologies associated with to maritime supply chain.

Author Response

Many thanks for your precious time and efforts expended in reviewing our paper. We attempted to address all the remaining uncertainties. Our hope is that the paper was further clarified.

We have addressed the first reviewer’s comments in the “Response_Reviewer_1” document. In addition, we have included the changes in the new version manuscript in blue color.

Reviewer 2 Report

[Comment 1] Novelty

[Subcomment 1a] The authors need to be clear on whether they propose the whole system from scratch without any reference. I could barely see any reference cited when presenting the details on the proposed systems. Please be clear on which part were developed by the authors and which were taken from any previous study (and please cite the study).

[Subcomment 1b] (lines 206-207) How about the cargo monitoring technology in land and air transportation? The authors need to whether the applied technology in the sea transportation is same with the ones used in land and air transportation. For any similar function(s), the authors must state how their proposed methodology outperforms the state-of-the-art methods used in land and air transportation. When providing the explanations, please relate the used technology for cold chain on land/air transportation as well.

[Subcomment 1c] (lines 369-370) The authors must provide a table comparing advantages and disadvantages of their proposed systems with the best existing one (please mark whether the compared system is used on sea, land, or air transportation).

[Subcomment 1d] Please add literature review about "Internet of Ships" to show how much research in this field has been emerging, especially related to the topic of this study.

[Comment 2] Proposed systems

[Subcomment 2a] The authors need to show how the proposed systems were tested.

[Subcomment 2b] Please provide more information about the project related to this paper to prove how well the authors have designed the systems.

[Comment 3] Writing quality

[Subcomment 3a] When citing a reference at the earlier part of any sentence, please mention the authors name, e.g., "Aslam et al. [17]", instead of only "[17]".

[Subcomment 3b] Please revise the grammatical errors and typing mistakes, e.g., in lines 262-266.

[Subcomment 3c] (lines 296-297) The terminologies are confusing. Please replace them, e.g., into 10 bays, 4 tiers, and 5 rows, following the terminologies used in https://link.springer.com/article/10.1007/s00291-003-0157-z. Please revise the whole manuscript.

[Subcomment 3d] I suggest revising "front end view" into "side view" in the figure as stated in the reference above as well.

[Subcomment 3e] Please revise some incompleted sentences, misused punctuation marks, misused capital letters, mistyped words, e.g. on lines 402, 414, 416, 431, 476, etc. I think the manuscript was not prepared well (there are too many writing errors).

Author Response

Many thanks for your precious time and efforts expended in reviewing our paper. We attempted to address
all the remaining uncertainties. Our hope is that the paper was further clarified.

We have addressed the first reviewer’s comments in the “Response_Reviewer_2” document. In addition, we have included the changes in the new version manuscript in blue color.

Round 2

Reviewer 2 Report

Thank you for your revisions.